# Amphiphilic Graft Copolymers as Templates for the Generation of Binary Metal Oxide Mesoporous Interfacial Layers for Solid-State Photovoltaic Cells

**DOI:** 10.3390/nano14040352

**Published:** 2024-02-13

**Authors:** Seung Man Lim, Hayeon Jeong, Juyoung Moon, Jung Tae Park

**Affiliations:** 1Department of Chemical Engineering, Konkuk University, 120 Neungdong-ro, Gwangjin-gu, Seoul 05029, Republic of Korea; 2Department of Chemical and Biomolecular Engineering, Yonsei University, 50 Yonsei-ro, Seodaemun-gu, Seoul 03722, Republic of Korea

**Keywords:** graft copolymer, mesoporous, metal oxide, interfacial layer, solid-state electrolyte, photovoltaic cells, dye-sensitized solar cells (DSSCs)

## Abstract

The binary metal oxide mesoporous interfacial layers (bi-MO meso IF layer) templated by a graft copolymer are synthesized between a fluorine-doped tin oxide (FTO) substrate and nanocrystalline TiO_2_ (nc-TiO_2_). Amphiphilic graft copolymers, Poly(epichlorohydrin)-*graft*-poly(styrene), PECH-*g*-PS, were used as a structure-directing agent, and the fabricated bi-MO meso IF layer exhibits good interconnectivity and high porosity. Even if the amount of ZnO in bi-MO meso IF layer increased, it was confirmed that the morphology and porosity of the bi-MO meso IF layer were well-maintained. In addtion, the bi-MO meso IF layer coated onto FTO substrates shows higher transmittance compared with a pristine FTO substrate and dense-TiO_2_/FTO, due to the reduced surface roughness of FTO. The overall conversion efficiency (*η*) of solid-state photovoltaic cells, dye-sensitized solar cells (DSSCs) fabricated with nc-TiO_2_ layer/bi-MO meso IF layer TZ1 used as a photoanode, reaches 5.0% at 100 mW cm^−2^, which is higher than that of DSSCs with an nc-TiO_2_ layer/dense-TiO_2_ layer (4.2%), resulting from enhanced light harvesting, good interconnectivity, and reduced interfacial resistance. The cell efficiency of the device did not change after 15 days, indicating that the bi-MO meso IF layer with solid-state electrolyte has improved electrode/electrolyte interface and electrochemical stability. Additionally, commercial scattering layer/nc-TiO_2_ layer/bi-MO meso IF layer TZ1 photoanode-fabricated solid-state photovoltaic cells (DSSCs) achieved an overall conversion efficiency (*η*) of 6.4% at 100 mW cm^−2^.

## 1. Introduction

As fossil fuels are depleting, scientists are looking for ways to replace these fuels with renewable energy sources. Solar energy is the most apparent renewable energy due to its abundance, unlimited supply, and lack of carbon dioxide emissions. Solar cells are classified into silicon solar cells, organic solar cells, and dye-sensitized solar cells. Among these, the use of dye-sensitized solar cells (DSSCs) is an attractive option to replace silicon solar cells; these solar cells are environmentally friendly and low-cost compared to silicon solar cells and have relatively high efficiency [1]. Liquid electrolytes with a triiodide/iodide redox pair are the most often utilized electrolyte in DSSCs. Although this type of electrolyte has outstanding photovoltaic performance, it has many drawbacks, including solvent evaporation, leakage, chemical degradation, counter electrode corrosion, dye absorption, and inadequate sealing of these cells in long-term applications. Another widely acknowledged idea to increase the operational stability of DSSC devices is to substitute the solid-state electrolyte for the liquid electrolyte.

A DSSC consists of a photoanode, a counter electrode, and electrolyte [2]. Among these components, the photoanode is the most important, as it is used to transfer dye-derived photoelectrons to external circuits. The ideal photoanode has a large surface area, high dye absorption, fast electron transfer rate, and suppressed electron recombination. TiO_2_ [3,4,5,6,7,8], SnO_2_ [9,10,11,12,13], and ZnO [14,15,16,17,18,19,20,21,22,23,24,25,26,27] are used as photoanodes. In general, TiO_2_ is used as a photoanode because it has an appropriate energy band gap, a large specific surface area, stable dye adsorption, and acts as a scattering layer leading to enhanced light harvesting. However, TiO_2_ has relatively low electron mobility; that is, electron recombination is induced. The more extensive the electron recombination, the lower the efficiency of the DSSC.

Alternatively, ZnO has an electron band gap similar to that of TiO_2_ but has high electron mobility [28,29]. There are many ZnO-based DSSCs with various morphologies, such as ZnO nanorods [14,15,16,17], nanoflowers [18,19,20], nanotubes [21,22,23], and ZnO core-shell structures [24,25,26,27]. However, ZnO-based DSSCs exhibit relatively low efficiencies compared to TiO_2_-based DSSCs because a complex that forms between ZnO and carboxylic acid in the dye decreases efficiency. Therefore, ZnO is not used alone as a photoanode but is mixed with other materials to improve the low stability while making use of its high electron mobility. A mesoporous TiO_2_ layer serves as the photoanode of DSSCs, and it should act as an appropriate structure to hold sensitizing dye sensitizers in place, allow sunlight to pass through the electrolyte, and transport electrons that produce light to the collector electrode. It is frequently composed of commercial TiO_2_ nanoparticles, which are typically 20 nm in size. By lowering the particle size, the surface area and dye absorption capacity are typically increased. However, surface imperfections that serve as electron traps can encourage electron–hole recombination, which will lower a nanoparticulate photoanode’s photocurrent density. In the meantime, a mesoporous TiO_2_ layer may accurately represent both dye-loading and light-collecting capabilities. In addition to having a high surface area from nanocrystalline aggregates, this structure allows the photoanode to harvest light since hierarchical particle sizes are similar to those in the visible region [30,31,32,33,34,35,36,37].

In this report, binary metal oxide mesoporous interfacial layers (bi-MO meso IF layer) templated by amphiphilic graft copolymer PECH-*g*-PS are synthesized between an FTO substrate and nc-TiO_2_ layer. Atom-transfer radical polymerization (ATRP) was used to create a PECH backbone, and PS (PECH-g-PS) was used as a structure-directing agent to create the bi-MO meso IF layer using a sol-gel method. Improved light harvesting, good interconnectivity, and reduced interfacial resistance were achieved by incorporating the bi-MO meso IF layer between the FTO substrate and nc-TiO_2_ layer. Additionally assessed were the photovoltaic characteristics of solid-state photovoltaic cells (DSSCs) based on the bi-MO meso IF layer, nc-TiO_2_ layer, and commercial scattering layer.

## 2. Experimental Section

### 2.1. Materials

Titanium (IV) isopropoxide (TTIP, 97%), zinc oxide (ZnO), poly(ethylene glycol) (PEG 10k) (Mn = 10,000 g mol^−1^), lithium iodide (LiI), 1-methyl-3-propylimidazolium iodide (MPII), iodine (I_2_), hydrogen chloride (HCl 37%), titanium diisopropoxide bis(acetylacetonate), chloroplatinic acid hexahydrate (H_2_PtCl_6_·6H_2_O), and acetonitrile were purchased from Aldrich. Tetrahydrofuran (>99.8%) was purchased from J. T. Baker. Commercial TiO_2_ paste (Dyesol paste, 18NR-T) was purchased from Dyesol. Ruthenium dye (N719) was purchased from Solaronix. FTO conductive glass (7 Ω) was purchased from Pilkington, France.

### 2.2. Synthesis of Amphiphilic Graft Copolymers, PECH-g-PS

The amphiphilic graft copolymers PECH-*g*-PS were used as structure-directing agents for the bi-MO meso IF layer, which was synthesized using the ATRP method, as described in previous research [38]. Briefly, 2 g of PECH was dissolved in 25 mL of toluene while stirring at 70 °C for 3 h. After cooling the solution to room temperature, 12 g of styrene, 0.16 g of CuCl, and 0.48 mL of hexamethyltriethylenetetramine (HMTETA) were added to the solution. The resulting green mixture was stirred until it became a homogeneous solution, then it was purged with nitrogen for 30 min. The mixture was placed in a 100 °C oil bath for 8 h. After polymerization, the resultant polymer solution was diluted with tetrahydrofuran (THF). After passing the solution through a column with activated Al_2_O_3_ to remove the catalyst, the solution was precipitated into methanol. The amphiphilic graft copolymers PECH-*g*-PS were obtained and dried in a vacuum oven overnight at room temperature.

### 2.3. Preparation of Binary Metal Oxide Mesoporous Interfacial Layer (bi-MO Meso IF Layer)

A series of samples was prepared by varying the amount of ZnO from 0 to 6 mg. First, 0.04 g of amphiphilic graft copolymers PECH-*g*-PS was dissolved in 1.5 mL of THF with vigorous stirring for 2 h. Separately, 150 μL of HCl and 150 μL of H_2_O were added to a 300 μL solution of TTIP sequentially, followed by vigorous stirring for 15 min. The TTIP/HCl/H_2_O solution was then added to the amphiphilic graft copolymers’ PECH-*g*-PS solution. Next, different amounts of ZnO (0, 2, 4, and 6 mg) were added to the solution and aged for 3 h with vigorous stirring at room temperature. Based on the amount of ZnO (0, 2, 4, and 6 mg), the bi-MO meso IF layers were named bi-MO meso IF layer TZ0, bi-MO meso IF layer TZ1, bi-MO meso IF layer TZ2, and bi-MO meso IF layer TZ3, respectively.

### 2.4. Preparation of Photoanode

Before depositing the nanocrystalline TiO_2_ layer (nc-TiO_2_ layer), cleaned FTO was coated with a bi-MO meso IF layer solution using a spin coater, followed by calcination at 450 °C for 30 min. Then, the nc-TiO_2_ layer was deposited onto the bi-MO meso IF layer using a doctor-blade method and dried at 50 °C followed by 80 °C for 1 h each. Finally, this photoanode was calcined at 450° C for 30 min.

### 2.5. Fabrication of DSSCs

DSSCs with an active area of 0.16 cm^−1^ were fabricated according to our previous method [39,40,41]. First, the FTO glass was sonicated with isopropanol and chloroform and dried in air. For the dense-TiO_2_ layer, titanium (IV) bis(ethyl acetoacetato) diisopropoxide in 1-butanol was spin coated onto FTO glass, followed by calcination at 450 °C for 30 min. Then, the nc-TiO_2_ layer was deposited onto the dense-TiO_2_ layer-coated FTO glass using a doctor-blade method and dried at 50 °C and 80 °C for 1 h each, and calcined at 450 °C for 30 min. This photoanode was immersed in a dye solution of 13 mg N719 in 50 mL of ethanol for 3 h at 50 °C. Subsequently, the resulting photoanode was washed several times with ethanol and dried in air. Solid-state electrolytes, containing LiI, I_2_, and PEG 10K in acetonitrile solvent, was cast onto the photoanode. A Pt counter electrode was prepared by spin coating 1 wt% H_2_PtCl_6_ in an isopropanol solution onto FTO glass and calcined at 450 °C for 30 min. Subsequently, the prepared photoanode and Pt counter electrode were superimposed and sealed with epoxy resin.

### 2.6. Characterization

Using an ATR facility, the FT-IR spectra of materials were obtained using an Excalibur Series FTIR (DIGLAB Co.) instrument, covering the frequency range of 4000 to 400 cm^−1^. XRD was used to identify the phase of specimens at 40 kV and 300 mA, using Rigaku CuKa radiation (λ = 1.5406 Å). In the 2 theta range of 5 to 60°, data were gathered using a step interval of 0.1° and a measuring period of 2 s per point. Surface and cross-sectional images were characterized by field emission scanning electron microscopy (FE-SEM, SU 8000, Hitachi). An aberration-corrected SEM, outfitted with an energy-dispersive X-ray spectrometer (EDS), operating at an acceleration voltage of 200 kV, was used to perform the detailed nanoarchitecture of the bi-MO meso IF layer. Transmittance spectra were obtained using UV-vis spectroscopy (MEGA 500, Scinco Co.) in the wavelength range of 300–800 nm. In order to measure photocurrent and voltage, a Keithley model 2400 source measuring unit was used. The light source was an ABE Technologies class A solar simulator (model 11,000) with a 1000 W xenon lamp (Oriel, 91193). A certified reference Si solar cell (Fraunhofer Institute for Solar Energy System, Mono-Si + KG filter, Certificate No. C-ISE269) was used to adjust the light intensity, to produce a sunlight intensity of one (100 mW/cm^2^). To stop more light from entering through the lateral space, a black mask with an opening was placed over the DSSCs during the photocurrent–voltage measurements. The computation of the photoelectrochemical performances was conducted using Equations (1) and (2):*FF = V_max_ · J _max_/Voc · Jsc*(1)
*η = V _max_ · J _max_/Pin · 100 = Voc · Jsc · FF/Pin · 100*(2)
which is composed of the following: *Pin* is the incident light power; *FF* is the fill factor; *η* is the overall energy conversion efficiency; *J _max_* (mA/cm^2^) and *V_max_* (V) are the current density and voltage in the J-V curve, respectively, at the point of maximum power output; and *Jsc* is the short-circuit current density (mAcm^2^). The EIS data were measured using a compactstat electrochemistry analyzer (IVIUM Technologies) with a frequency range of 0.01 Hz to 0.1 MHz and potential modulation of 0.2 V. Incident photo to current conversion efficiency (IPCE) spectra were acquired with a 300 W xenon lamp and a monochromator, equipped with order sorting filters (K3100) at a spectral resolution of 5 nm. The IPCE value was calculated using the following formula:*IPCE = h c I/λ p*(3)
where *h* and *c* stand for the terms Planck’s constant and the speed of light in a vacuum, respectively. *I* is the photocurrent density (mA/cm^2^), and *λ* and *p* are the wavelength (nm) and intensity (mA/cm^2^) of the incident monochromatic light, respectively. Under AM 1.5 (100 mW/cm^2^) light illumination, the EIS data were measured using a compactstat electrochemistry analyzer (IVIUM Technologies) with a frequency range of 0.01 Hz to 0.1 MHz and potential modulation of 0.2 V.

### 2.7. Measurement of Dye Adsorption Value

The dye-sensitized photoanode was dipped into 5 mL of a 0.01 M NaOH solution in ethanol-H_2_O (1:1). The mixture was stirred until the complete desorption of the dye. This mixture solution was measured by UV-Vis spectroscopy at 515 nm in absorption mode. Absorbance at 515 nm was used to calculate the amount of adsorbed N719 dye molecules, according to the Beer–Lambert law,
A = *εlc*(4)
where A is the absorbance of the UV–visible spectra at 515 nm, *ε* = 14,100/M cm is the molar extinction coefficient of the dye at 515 nm, *l* is the path length of the light beam, and *c* is the dye concentration [42].

## 3. Results and Discussion

The preparation of binary metal oxide mesoporous interfacial layers (bi-MO meso IF layer) templated by amphiphilic graft copolymers, Poly(epichlorohydrin)-graft-poly(styrene) and PECH-*g*-PS, is illustrated in Figure 1. The templating amphiphilic graft copolymer, PECH-*g*-PS, which is composed of a hydrophilic PECH backbone chain and hydrophobic PS side chain, was used as a structure-directing agent in the sol-gel reaction. The hydrophilic PECH backbone undergoes a Ti precursor and ZnO NP selective reaction to form a self-assembled micelle. The PECH-*g*-PS/Ti precursor and ZnO NP hybrid phase splits into nanometer-sized domains upon calcination at 450 °C. The Ti precursor and ZnO NP sits in the hydrophilic PECH domains to form a bi-MO meso IF layer around the hydrophobic PS areas that produce pores. The chemical method for the ATRP synthesis of the amphiphilic graft copolymers PECH-*g*-PS employing direct initiation of the PECH chlorine atoms is shown in Appendix A. The ATRP-based “grafting from” approach is a popular and effective polymerization technique for creating functional sol-gel templates [30]. The FT-IR spectra of the amphiphilic graft copolymers, PECH-*g*-PS and pristine PECH, are shown in Appendix A. The ether stretching band on the hydrophilic backbone chain of the pristine PECH is responsible for the strong absorption bands that were seen at about 1102 cm^−1^. Three absorption bands at 1455, 1499, and 1600 cm^−1^ emerged during PS graft polymerization; these bands were ascribed to C=C stretching vibrations of PS. The hydrophobic PS side chain was successfully grafted using ATRP from the chlorine atoms on the hydrophilic PECH backbone, as evidenced by the FT-IR spectroscopic data.

Figure 1 displays the XRD pattern of the bi-MO meso IF layer on FTO that was created using the amphiphilic graft copolymers’ PECH-*g*-PS template. A number of strong peaks, centered at 2 theta = 25.5, 37.9, and 54.3, were seen in the bi-MO meso IF layer TZ0, TZ1, TZ2, and TZ3; these correspond to the (101), (004), and (105) reflections of the anatase form of TiO_2_, respectively [43]. The (110), (101), (200), and (211) crystal planes of the rutile SnO_2_ phase of the FTO substrate are represented by a number of strong peaks at 2 theta values of 26.8, 33.9, 38.1, and 51.6, respectively [44]. Furthermore, small diffraction peaks identified as (100), (101), and (110) reflections of the hexagonal wurtzite structure of ZnO were seen in the XRD pattern for the bi-MO meso IF layers TZ1, TZ2, and TZ3 on the FTO glass substrate [45]. Overall, Figure 1 shows that, according to the instrumental analysis for material characterization, the bi-MO meso IF layers were successfully prepared on the FTO substrate.

Figure 2 displays the surface and cross-sectional FE-SEM images of the double-layer structure associated with the nc-TiO_2_ layer and bi-MO meso IF layer TZ1 on the FTO substrate. A uniformly distributed mesoporous structure with high porosity can be seen in Figure 2a, which indicates that a higher surface area was achieved in the bi-MO meso IF layer TZ1, leading to a higher dye adsorption value in solid-state photovoltaic cells. In Figure 2a,b, the surface pore size and cross-sectional thickness of the bi-MO meso IF layer TZ1 were found to be approximately 30~50 nm and 500 nm, respectively. FE-SEM images of a bi-MO meso IF layer TZ1 with a robust attachment structure to the FTO substrate and nc-TiO2 layer is displayed in Figure 2c,d, which results in improved electron transfer and suppressed electron recombination in solid-state photovoltaic cells. Ti and Zn ions are uniformly deposited on the bi-MO meso IF layer TZ1, according to the results of energy dispersive X-ray spectroscopy (EDS) mapping pictures displayed in Appendix A. Also, it was shown that the shape, porosity, and metal ion distribution of the bi-MO meso IF layer were preserved, even when the concentration of ZnO in the layer varied, as shown in Appendix A.

As seen in Figure 3, the transmittance spectra of the bi-MO meso IF layer on FTO were acquired in the wavelength range of 300–800 nm, in which the bi-MO meso IF layer exhibited higher transmittance compared with the pristine FTO and dense TiO_2_ layer on FTO. This is most likely caused by the bi-MO meso IF layer that formed on the FTO substrate, which improved the uniformity of the film surface and ultimately increased the light transmittance of the device. When light passes through a medium with a different refractive index vertically, the reflectance *R* can be expressed as the following equation:*R* = [(n_1_ − n_2_)/(n_1_ + n_2_)]^2^(5)
where n_1_ and n_2_ represent the refractive index of each medium. The constructive and destructive interferences are expressed as the following equations:*2nd* = (*m* + *1/2*) *λ* (constructive interference)(6)
*2nd* = *mλ* (destructive interference)(7)
where *n* and *d* represent the refractive index and the thickness of medium, respectively. *λ* represents the wavelength of the light and *m* is a constant (m = 1, 2, 3…). According to Equation (5), reflection occurred at the air/bi-MO meso IF layer and bi-MO meso IF layer/FTO glass interfaces and was proportional to the difference in refractive index. According to Equations (6) and (7), constructive and destructive interferences occurred, depending only on the thickness and refractive index of the bi-MO meso IF layer, as seen in Figure 3, because the thickness and refractive index of the FTO glass are unchanged. The refractive index of the bi-MO meso IF layer could be influenced by the relative amount of pristine TiO_2_ (*n* = 2.6) and ZnO (*n* = 2.0) [46,47]. The lower refractive index of the bi-MO meso IF layer was observed as the amount of ZnO was increased, indicating that the porosity of the bi-MO meso IF layer influenced the refractive index.

The current density–voltage (*J–V*) curves of solid-state photovoltaic cells (DSSCs) with the bi-MO meso IF layer, templated by amphiphilic graft copolymers PECH-*g*-PS, were performed at 100 mW/cm^2^, as seen in Figure 4a, and cell performance and related parameters, including short-circuit current density (*J_SC_*), open-circuit voltage (*V_OC_*), fill factor (FF), overall conversion efficiency (*η*), and dye adsorption amount, are summarized in Table 1. The overall conversion efficiency of the bi-MO meso IF layer used as the photoanode was higher than that of the nc-TiO_2_ layer/dense-TiO_2_ layer, indicating that the nc-TiO_2_ layer/bi-MO meso IF layer TZ1 achieved the highest value (5.0%). A higher *J_SC_* was observed in solid-state photovoltaic cells (DSSCs) with the bi-MO meso IF layer TZ1 than those with the dense-TiO_2_ layer. *J_SC_* is a function of dye adsorption value and light harvesting; therefore, the bi-MO meso IF layer TZ1, with high porosity, exhibited high dye adsorption (70 nmol/cm^2^), and higher light harvesting, which resulted in a higher *J_SC_*, as confirmed in the IPCE results (Figure 4c). Note that the dye adsorption value of the nc-TiO_2_ layer/dense-TiO_2_ layer is 55 nmol/cm^2^ (Table 1, Appendix A). Also, the higher electron mobility of the ZnO bi-MO meso IF layer TZ1 enhanced the *J_SC_* and overall conversion efficiency compared to the solid-state photovoltaic cells (DSSCs) with the bi-MO meso IF layer TZ0. Interfacial and internal resistance of the solid-state photovoltaic cells (DSSCs) with the bi-MO meso IF layer were investigated using electrochemical impedance spectroscopy (EIS) analysis under 100 mW/cm^2^, as shown in Figure 4b; the resulting equivalent circuit is shown in Appendix A. The electrochemical parameters of the solid-state photovoltaic cells (DSSCs) with bi-MO meso IF layers are listed in Table 2. A frequency range from 0.1 Hz to 0.1 MHz with an AC amplitude of 0.01 V at the *V_OC_* was used. The parameters related to resistance consisted of the ohmic series resistance of FTO glass (*R_s_*), charge–transfer resistance at the counter electrode–electrolyte interface (*R_1_*), charge–transfer resistance at the photoanode–electrolyte interface (*R_2_*), and the Warburg diffusion resistance of the redox I^−^/I_3_^−^ couple in electrolyte (W_s_). Among these parameters, the *R_2_* value is important in order to investigate how effectively the bi-MO meso IF layer in solid-state photovoltaic cells (DSSCs) worked electrochemically. Solid-state photovoltaic cells (DSSCs) with the nc-TiO_2_ layer/bi-MO meso IF layer TZ1 (9.6 Ω) as the photoanode had a smaller *R_2_* value than those of the other solid-state photovoltaic cells (DSSCs), indicating a smaller charge–transfer resistance between the photoanode and electrolyte, and higher *FF* and conversion efficiency. Recombination resistance (*R_rec_*) at the photoanode–electrolyte interface was also measured using EIS analysis under dark conditions, as shown in Appendix A. Measurement was performed at −550 mV, and the frequency range and AC amplitude conditions were the same as the EIS in bright conditions. Solid-state photovoltaic cells (DSSCs) with the nc-TiO_2_ layer/bi-MO meso IF layer exhibited smaller *R_rec_* values, demonstrating a higher electron recombination rate. This demonstrated that the electron recombination was suppressed by high electron mobility with good connectivity between the FTO, nc-TiO_2_ layer, and bi-MO meso IF layer, which helped enhance the *FF* value. As shown in Figure 4b, the cell performance of solid-state photovoltaic cells (DSSCs) with the nc-TiO_2_ layer/bi-MO meso IF layer remained unchanged over 15 days, suggesting that the bi-MO meso IF layer with solid-state electrolytes has enhanced electrochemical stability at the electrode/electrolyte interface. Also, commercial scattering layer/nc-TiO_2_ layer/bi-MO meso IF layer TZ1 photoanode-prepared solid-state photovoltaic cells (DSSCs) achieved an overall conversion efficiency (*η*) of 6.4% at 100 mW cm^−2^, as shown in Appendix A**.** As indicated in Appendix A, it should be noted that the efficiency of solid-state photovoltaic cells (DSSCs) with the nc-TiO_2_ layer/bi-MO meso IF layer is among the highest values recorded for solid-state photovoltaic cells with a mesoporous layer to date [48,49,50,51,52,53,54].

## 4. Conclusions

We reported the bi-MO meso IF layer templated by amphiphilic graft copolymers (PECH-*g*-PS) prepared by a sol-gel method, with high porosity and good interconnectivity. Solid-state photovoltaic cells (DSSCs) fabricated with an nc-TiO_2_ layer/bi-MO meso IF layer TZ1 exhibited a higher overall conversion efficiency (5.0%) than those with an nc-TiO_2_ layer/bi-MO meso IF layer TZ0 (4.7%) and nc-TiO_2_ layer/dense-TiO_2_ layer (4.2%) at 100 mW/cm^2^, demonstrating an improved electron transport rate and reduced interfacial resistance with suppressed recombination resistance. Furthermore, solid-state photovoltaic cells (DSSCs) fabricated with a commercial scattering layer, nc-TiO_2_ layer, and bi-MO meso IF layer photoanode had an overall conversion efficiency (*η*) of 6.4% at 100 mW cm^−2^. This research presents future options for the development of renewable energy devices and emphasizes the potential of utilizing a bi-MO meso IF layer for the development of efficient solid-state photovoltaic cells (DSSCs) with enhanced efficiency and long-term stability.

## Data Availability

The original contributions presented in the study are included in the article and Appendix A, further inquiries can be directed to the corresponding author.

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
