# Peer review of "Amphiphilic Graft Copolymers as Templates for the Generation of Binary Metal Oxide Mesoporous Interfacial Layers for Solid-State Photovoltaic Cells"

_nanomaterials, 2024, doi:10.3390/nano14040352_

Round 1

Reviewer 1 Report

Comments and Suggestions for Authors

The authors presented a method for producing a mesoporous TiO2 layer, for solid state dye-sensitized solar cells. There are several issues, in particular with relation to literature review and benchmarking.

1. Traditional liquid DSSCs, organic solar cells, and also perovskite solar cells have all exhibited much higher solar cell efficiencies. The authors should provide some motivation in the introduction why solid state DSSCs are worth pursuing. 

2. There are also a lot of literature describing the preparation of mesoporous TiO2. The authors should do a proper literature review, and describe what is novel or interesting with their proposed method. 

3. The authors claim that the higher surface area TiO2 should lead to greater dye adsorption. They should back this up with UV-Vis absorption spectra of the dye/mesoporous TiO2 and dye/dense TiO2 layers. 

Author Response

Thank you very much for your careful review on our manuscript. We revised the manuscript mostly as suggested:

Reviewer # 1

< General Comments >

The authors presented a method for producing a mesoporous TiO2 layer, for solid state dye-sensitized solar cells. There are several issues, in particular with relation to literature review and benchmarking.

< Specific Comments >

(1) Traditional liquid DSSCs, organic solar cells, and also perovskite solar cells have all exhibited much higher solar cell efficiencies. The authors should provide some motivation in the introduction why solid state DSSCs are worth pursuing.

< Response >

To address this comment, we added the following descriptions.

“Solar cells are classified into silicon solar cells, organic solar cells, perovskite solar cells and dye-sensitized solar cells. Among these, the use of dye-sensitized solar cells (DSSCs) is an attractive option to replace silicon solar cells; these solar cells are environmentally friendly and low-cost compared to silicon solar cells, and have a relatively high efficiency. [1] Liquid electrolyte with a triiodide/iodide redox pair is the most often utilized electrolyte in DSSCs. Although this type of electrolyte has outstanding photovoltaic performance, it has many drawbacks, including solvent evaporation and leakage, chemical degradation, counter electrode corrosion, dye absorption, and inadequate sealing of these cells in long-term applications. Another widely acknowledged idea to increase the operational stability of DSSC devices is to substitute solid-state electrolyte for the liquid electrolyte.”

(2) There are also a lot of literature describing the preparation of mesoporous TiO2. The authors should do a proper literature review, and describe what is novel or interesting with their proposed method.

< Response >

To address this comment, we added the following descriptions and cited the paper as Ref # 30~37.

“A mesoporous TiO2 layer serves as the photoanode of DSSCs, and it should act as an appropriate structure to hold sensitizing dye sensitizers in place, allow sunlight to pass through the electrolyte, and transport electrons that produce light to the collector electrode. It is frequently composed of commercial TiO2 nanoparticles, which are typically 20 nm in size. By lowering the particle size, the surface area and dye absorption capacity are typically increased. However, surface imperfections that serve as electron traps will encourage electron–hole recombination, which will lower a nanoparticulate photoanode's photocurrent density. In the meantime, a mesoporous TiO2 layer might accurately represent both the dye-loading and light collecting capabilities. In addition to having a high surface area from the nanocrystalline aggregates, this structure allows the photoanode to harvest light since the hierarchical particle sizes are similar to those of the visible region. [30-37]”

  1. Cao, D.; Wang, A.; Yu, X.; Yin, H.; Zhang, J.; Mi, B.; Gao, Z. Room-temperature preparation of TiO2/graphene composite photoanodes for efficient dye-sensitized solar cells, J. Colloid Interface Sci. 2021, 586, 326.
  2. Yyagi, J.; Gupta, H.; Purohit, L.P. Mesoporous ZnO/TiO2 photoanodes for quantum dot sensitized solar cell, Opt. Mater. 2021, 115, 111014.
  3. Buapuean, T.; Jarudilokkul, S. Synthesis of mesoporous TiO2 with colloidal gas aphrons, colloidal liquid aphrons, and colloidal emulsion aphrons for dye-sensitized solar cells, Mater. Today Chem. 2020, 16, 100235.
  4. Yan, H.; Chen, M.; Liu, W.; Wang, P.; Liu, M.; Liu, Y.; Ye, L.; Gu, M. Ti3C2 MXene quantum dots decorated mesoporous TiO2/Nb2O5 functional photoanode for dye-sensitized solar cells, Opt. Mater. 2023, 140, 113902.
  5. Graddage, N.; Ouyang, J.; Lu, J.; Chu, T.-Y.; Zhang, Y.; Li, Z.; Wu, X.; Malenfant, P. R. L.; Tao, Y. Near-Infrared-II Photodetectors Based on Silver Selenide QuantumDots on Mesoporous TiO2 Scaffolds, ACS Appl. Nano Mater. 2020, 3, 12209.
  6. Zhang, L.; Zhang, J.; Tang, X.; Chen, Y.; Wang, X.; Deng, Z.; Wang, C.; Yang, X.; Sun, B. Densely Packed D−π–A Photosensitizers on TiO2 Enable Efficient Dye-Sensitized Solar Cells, ACS Appl. Energy Mater.2023, 6, 4229.
  7. Wu, T.; Deng, G.; Zhen, C. Metal oxide mesocrystals and mesoporous single crystals: synthesis, properties and applications in solar energy conversion, J. Mater. Sci. Technol. 2021, 73, 9.
  8. Cao, D.; Yin, H.; Yu, X.; Zhang, J.; Jiao, Y.; Zheng, W.; Mi, B.; Gao, Z. Role of Modifying Photoanodes by Organic Titanium on Charge Collection Efficiency Enhancement in Dye-Sensitized Solar Cells, Adv. Eng. Mater. 2020, 22, 1901071.

(3) The authors claim that the higher surface area TiO2 should lead to greater dye adsorption. They should back this up with UV-Vis absorption spectra of the dye/mesoporous TiO2 and dye/dense TiO2 layers.

< Response >

To address this comment, we added the following descriptions and added the Table 1 and F“JSC is a function of dye adsorption value and light harvesting; therefore, with bi-MO meso IF layer TZ1with high porosity exhibited high dye adsorption (70 nmol/cm2), and higher light harvesting which resulted in higher JSC, as confirmed in IPCE results. (Figure 4c) Note that the dye adsorption value of nc-TiO2 layer/dense-TiO2 layer is 55 nmol/cm2. (Table 1, Figure S7)”

Reviewer 2 Report

Comments and Suggestions for Authors

This manuscript by Lim et al. describes the fabrication of binary metal oxide layers onto fluorine-doped tin oxide support that enhances charge separation. The results improve the interfacial charge transfer rate and electrochemical stability of the device. In the study, amphiphilic graft copolymers are used as a structure-directing agent. It allows the formation of uniform porosity that allows high uptake of dyes. The report is technically sound. There are unfortunately some minor issues that require the author’s attention. 

(1) In the Experimental Section, the details of the preparation of bi-Mo meso IF layer should be provided.

(2) P.210, it should be 500 nm for the cross-sectional thickness of the bi-MO meso IF layer TZ1, instead of 500 mm.

(3) Comparison of other solid state DSSC should be made.

Comments on the Quality of English Language

fine.

Author Response

Thank you very much for your careful review on our manuscript. We revised the manuscript mostly as suggested:

Reviewer # 2

< General Comments >

This manuscript by Lim et al. describes the fabrication of binary metal oxide layers onto fluorine-doped tin oxide support that enhances charge separation. The results improve the interfacial charge transfer rate and electrochemical stability of the device. In the study, amphiphilic graft copolymers are used as a structure-directing agent. It allows the formation of uniform porosity that allows high uptake of dyes. The report is technically sound. There are unfortunately some minor issues that require the author’s attention.

< Specific Comments >

(1) In the Experimental Section, the details of the preparation of bi-Mo meso IF layer should be provided.

< Response >

To address this comment, we added the following descriptions.

“2.2. Synthesis of Amphiphilic Graft Copolymers, PECH-g-PS

The amphiphilic graft copolymers, PECH-g-PS was used as a structure direct-ing-agent for bi-MO meso IF layer, which was synthesized using the ATRP method, as described in previous research. [38] Briefly, 2 g of PECH was dissolved in 25 mL of toluene while stirring at 70 °C for 3 h. After cooling the solution to room temperature, 12 g of sty-rene, 0.16 g of CuCl and 0.48 mL of hexamethyltriethylenetetramine (HMTETA) were added to the solution. The resulting green mixture was stirred until it became a homogeneous solution, then it was purged with nitrogen for 30 min. The mixture was placed in a 100 oC oil bath for 8 h. After polymerization, the resultant polymer solution was diluted with tetrahydrofuran (THF). After passing the solution through a column with activated Al2O3 to remove the catalyst, the solution was precipitated into methanol. The amphiphilic graft copolymers, PECH-g-PS was obtained and dried in a vacuum oven overnight at room temperature.

2.3. Preparation of Binary Metal Oxide Mesoporous Interfacial Layer (bi-MO Meso IF Layer)

A series of samples was prepared by varying the amount of ZnO from 0~6 mg. First, 0.04 g of amphiphilic graft copolymers, PECH-g-PS was dissolved in 1.5 mL of THF with vigorous stirring for 2 h. Separately, 150 μL of HCl and 150 μL of H2O were added to a 300 μL solution of TTIP sequentially, followed by vigorous stirring for 15 min. The TTIP/HCl/H2O solution was then added to the amphiphilic graft copolymers, PECH-g-PS solution. Next, different amounts of ZnO (0, 2, 4, and 6 mg) were added to the solution and aged for 3 h with vigorous stirring at room temperature. Based on the amount of ZnO (0, 2, 4, and 6 mg), the bi-MO meso IF layer were named bi-MO meso IF layer TZ0, bi-MO meso IF layer TZ1, bi-MO meso IF layer TZ2, and bi-MO meso IF layer TZ3, respectively.

2.4. Preparation of Photoanode

Before depositing nanocrystalline TiO2 layer (nc-TiO2 layer), cleaned FTO was coated with a bi-MO meso IF layer solution using a spin coater, followed by calcination at 450 °C for 30 min. Then, nc-TiO2 layer was deposited onto the bi-MO meso IF layer using a doctor blade method and dried at 50 °C followed by 80 °C for 1 h each. Finally, this photoanode was calcined at 450° C for 30 min.”

(2) P.210, it should be 500 nm for the cross-sectional thickness of the bi-MO meso IF layer TZ1, instead of 500 mm.

< Response >

To address this comment, we modified the following descriptions.

“In Figure 2a and b, the surface pore size and cross-sectional thickness of the bi-MO meso IF layer TZ1 were found to be approximately 30~50 nm and 500 nm, respectively.”

(3) Comparison of other solid state DSSC should be made.

< Response >

To address this comment, we added the following descriptions, the Table S1 and cited the paper as Ref # 48~54.

“As indicated in Table S1, it should be noted that the efficiency of the solid state photovoltaic cells (DSSC) with nc-TiO2 layer/bi-MO meso IF layer is among the highest values rec-orded for solid state photovoltaic cells with mesoporous layer to date. [48-54]”

  1. Mohsenzadegan, N.; Nouri, E.; Mohammadi, M. R. Efficient quasi-solid-state dye-sensitized solar cells aided by mesoporous TiO2 beads and a non-volatile gel polymer electrolyte, CrystEngComm. 2023, 25, 3210.
  2. Wen, J.; Liu, Y.; Li, T.; Liu, C.; Wang, T.; Liu, Y.; Zhou, Y.; Li, G.; Sun, Z. Low Cost and Strongly Adsorbed Melamine Formaldehyde Sponge Electrolyte for Nontraditional Quasi-Solid Dye-Sensitized Solar Cells, ACS Appl. Energy Mater. 2023, 6, 4952.
  3. Fang, D.; Tan, Y.; Ren, Y.; Zheng, S.; Xiong, F.; Wang, A.; Chang, K.; Mi, B.; Cao, D.; Gao, Z. Simple Solution Preparation of Cs2SnI6 Films and Their Applications in Solid-State DSSCs, ACS Appl. Mater. Interfaces. 2023, 15, 32538.
  4. Moon, J.; Shin, W.; Park, J. T.; Jang, H. Solid-state solar energy conversion from WO3 nano and microstructures with charge transportation and light scattering characteristics, Nanomaterials 2019, 9, 1797.
  5. Lim, S. M.; Moon, J.; Choi, G.H.; Baek, U.C.; Lim, J.H.; Park, J. T.; Kim, J. H. Surface carbon shell-functionalized ZrO2 as nanofiller in polymer gel electrolytes-based dye-sensitized solar cells, Nanomaterials 2019, 9, 1418.
  6. Lee, J.Y.; Choi, G.H.; Moon, J.; Choi, W.S.; Park, J.T. 1D Co4S3 Nanoneedle Array with Mesoporous Carbon Derived from Double Comb Copolymer as an Efficient Solar Conversion Catalyst, Appl. Surf. Sci. 2021, 535, 147637.
  7. Song, E.; Moon, J.; Lee, J.Y.; Lee, C.O.; Chi, W.S.; Park, J.T. High-Voltage Solar Energy Conversion Based on a ZIF-67 Derived Binary Redox-Quasi-Solid-State Electrolyte, J. Electroanal. Chem. 2021, 893, 115264.

Round 2

Reviewer 1 Report

Comments and Suggestions for Authors

The manuscript has improved after revision, and I am willing to accept it.